# Low-Cost Raman Spectroscopy Setup Combined with a Machine Learning Model

**DOI:** 10.3390/s25030659

**Published:** 2025-01-23

**Authors:** Catarina Domingos, Alessandro Fantoni, Miguel Fernandes, Jorge Fidalgo, Sofia Azeredo Pereira

**Affiliations:** 1Department of Electronics, Telecommunication and Computers, Lisbon School of Engineering (ISEL), Polytechnic University of Lisbon (IPL), Rua Conselheiro Emídio Navarro, n°1, 1959-007 Lisbon, Portugal; a47412@alunos.isel.pt (C.D.); mfernandes@deetc.isel.ipl.pt (M.F.); jorge.fidalgo@isel.pt (J.F.); 2Center of Technology and Systems (UNINOVA-CTS) and Associated Lab of Intelligent Systems (LASI), 2829-516 Caparica, Portugal; 3iNOVA4Health, NOVA Medical School, Faculdade de Ciências Médicas, NMS, FCM, Universidade NOVA de Lisboa, 1169-056 Lisbon, Portugal; sofia.pereira@nms.unl.pt

**Keywords:** sensor, point of care, Raman spectroscopy, instrumentation, diagnosis, kidney disease

## Abstract

The diagnosis of kidney diseases presents significant challenges, including the reliance on variable and unstable biomarkers and the necessity for complex and expensive laboratory tests. Raman spectroscopy emerges as a promising technique for analyzing complex fluids, like urine, and detecting important disease biomarkers. However, its complexity, high cost and limited accessibility outside clinical contexts complicate its application. Moreover, the analysis of Raman spectra is a challenging and intensive task. In response to these challenges, in this study, we developed a portable, simplified and low-cost Raman system designed to acquire high-quality spectra of liquid complex samples. Using the “Starter Edition” methodology from the OpenRAMAN project, the system was optimized through laser temperature adjustments, by evaluating the laser emission spectrum under different temperatures with a spectrometer, and through adjustment of the acquisition parameters of the software used, by acquiring the ethanol spectra. The system validation was performed through the acquisition of Raman spectra from five urine samples, demonstrating its consistency and sensitivity to composition variations in urine samples. Additionally, a neural network was designed and trained using methanol and ethanol solutions. The model’s hyperparameters were optimized to maximize its precision and accuracy, achieving 99.19% accuracy and 99.21% precision, with a training time of approximately 3 min, underlining the model’s potential for classifying simple Raman spectra. While further system validation with more samples, a more in-depth analysis of the biomarkers present in urine and the integration with more sophisticated elements are necessary, this approach demonstrates the system characteristics of affordability and portability, making it a suitable solution for point-of-care applications and offering simplified accessibility for assessing the diseases risk outside clinical contexts.

## 1. Introduction

The term acute kidney injury (AKI) reflects a set of heterogeneous conditions, leading to an abrupt decline in renal excretory function, causing azotemia and alterations in urinary flow. AKI is an increasing global concern and a costly clinical syndrome, affecting nearly one-quarter of all hospitalized patients worldwide [1].

Serum creatinine (sCr) levels and the volume of urine production, known as diuresis, are commonly used for the diagnosis of acute kidney injury (AKI); however, these have limitations. Serum creatinine is a biomarker with poor performance in detecting AKI since creatinine levels only increase when around half of baseline renal function has been lost. At this stage, AKI has already progressed to Chronic Kidney Disease (CKD) and the diagnosis is considered late. In addition, creatinine concentration fluctuates with muscle metabolism and with variations in extracellular volume, two very instable parameters in people with AKI. Similarly, urine production is influenced by several factors including the volume of fluids ingested and the use of diuretic medications [2,3,4]. Therefore, reliance solely on changes in the serum creatinine level results in a delay in AKI management, leading to unfavorable outcomes for patients [5].

The classical biomarker paradigm is that one test detects one disease; however, AKI is a complex disease with multiple causes, and one biomarker is not sufficient for early diagnosis. Thus, a panel of biomarkers may be necessary for the correct detection of AKI [6]; however, as most of the AKI biomarkers are measured by ELISA tests, such parallel simultaneous screening is not possible in a typical clinical procedure.

The classification of renal malfunctioning and its correlation with other diseases is of overwhelming importance for assessing the risk of kidney failure, addressing a safe lifestyle and allowing a gentle and safe aging process. Moreover, there are some specific situations, like hospital emergency departments, extreme sports competitions, or military operations, where it is not possible to send collected urine to a laboratory for a standard analysis. There is a clear need and strong request for a fast screening of urine to support the decision of the medical doctor about the need for immediate medical onsite care. Taking advantage of the recent technological advances and affordable commercial devices, optoelectronic and spectral analysis of urine can be performed onsite by low-cost portable detection technology interfaced with signal processing and machine learning algorithms. These algorithms, trained on a set of previously known diagnostic outputs, can correlate the urine optoelectronic measured parameters with the person’s health status [7].

Raman spectroscopy is presently proposed as a novel approach to a multiplexed measurement of specific AKI biomarkers and has become an increasingly valuable and comprehensive tool. This technique stands out for not requiring any prior sample preparation and its compatibility with complex and aqueous samples; since the Raman signal produced by water is relatively weak, it does not interfere with the signal of the other components. In addition, due to its non-destructive nature, it can be used as a diagnostic technique and in situ measurements. Thus, Raman spectroscopy allows for a complete characterization of samples, including complex biological matrices like urine, which consolidates the relevant role of this technology in clinical analysis and in the field of biomedicine [8], offering several advantages over conventional methods for detecting kidney diseases.

Raman spectroscopy reduces the time required for evaluation and provides comprehensive information on the urine constituents, enabling early detection of molecular markers associated with kidney diseases. This is accomplished by using only a small amount of sample and reducing the need for exhaustive laboratory tests [9].

Due to the low intensity of the Raman light scattered by molecules, one of the main challenges of this technique is the proper detection of its signal and obtaining spectra with analyzable peaks. The Raman system must be able to eliminate the intense Rayleigh radiation while amplifying the weak Raman radiation. To achieve this, the excitation radiation needs to have high power, be as monochromatic as possible and have high coherence and stability [10]. Currently, lasers are considered the ideal radiation source for Raman spectroscopy experiments as they emit highly collimated and directional beams, with high power and coherence, both spatially and temporally [11].

The fluorescence emitted by a sample, or by the impurities it may contain, is one of the aspects that require particular attention in Raman spectroscopy. If the analyzed material interacts with the incident radiation and emits fluorescence, the bands associated with this type of radiation can overlap the sample’s Raman spectrum. Fluorescence is significantly more intense than Raman radiation, resulting in high-intensity bands that can contaminate the acquired spectrum and complicate the detection of Raman peaks. To minimize fluorescence interference, a preliminary study should be carried out to analyze the sample’s fluorescence response to various wavelengths. This approach allows for the identification of a spectral region with minimal fluorescence emission, which corresponds to the optimal wavelength of that sample [12].

The laser’s emission range, also referred to as its spectral amplitude, also plays an important role in the spectral resolution of the Raman spectroscopy technique. When the laser’s emission range covers a wide range of wavelengths, the Raman spectral resolution is compromised, making it difficult to interpret the results and correctly assign the vibrational modes of the molecules present in the sample. Therefore, the laser chosen needs to have a wavelength that minimizes the fluorescence emitted by the sample and needs to be as monochromatic and stable as possible.

Raman spectrum analysis can quickly become difficult to interpret due to its complex nature and the large amount of information it contains, leading to the risk of losing important data and underseeing subtle concentration variations in biological markers. For this reason, automating the process of urine spectra interpretation appears to be a viable solution for their effective analysis and classification. A supervised learning model, based on a neural network, could recognize characteristic patterns in these spectra, aiding in the future possibility of diagnosis. If the supervised model is correctly trained with sufficient data and information, it can reduce the analysis time and ensure greater consistency in the results.

The primary objective of this study is to develop a portable and low-cost Raman spectroscopy-based system, capable of acquiring high-quality spectra, for potential future diagnosis applications. Furthermore, we aim to evaluate the possibility of using a supervised learning model to classify the Raman spectra acquired by the developed system. While the goal is to support kidney disease diagnostics, the versatility of the proposed system and the machine learning model extends its applicability to a wide range of diseases and biological samples. Once validated, this system could serve as a foundation tool for diagnosing other conditions where biochemical alterations in biofluids play a crucial role. Additionally, its affordability and portability make this system suitable for use in both clinical and emergency settings, where complex laboratory analyses are unfeasible.

By proposing a concurrent development of the hardware and signal processing architecture, this prototype attempts to move the system complexity from the hardware layer to the software layer. It targets the development of a Cyber–Physical System that can assist in overcoming the physical limitations of health services access, allowing health monitoring independently based on geographic location and economical condition.

## 2. Raman System Development

The methodology adopted for the development of the Raman system was based on the modular Starter Edition, available on the OpenRAMAN website, with the necessary adaptations to meet the specific needs of this project. The Starter Edition spectrometer was chosen because it is the simplest and most economical edition suitable to achieve the low-cost criterion of the system we intend to build [13].

### 2.1. Description of System Components

For simplicity’s sake, the system will be divided into its main components: radiation source, optical elements and detector. In addition, we will refer to the data collection software used. All the other important components can be found on the OpenRAMAN site.

Most of the components used were purchased from the supplier Thorlabs. However, some parts of the system needed to be modified or designed from scratch to properly fit the specifics of the system. In Figure 1, we present the optical layout of the system developed [14].

#### 2.1.1. Radiation Source

The main factors to take into consideration in laser selection are wavelength and emission range. To identify the optimal wavelength for Raman analysis, urine’s fluorescence emission was quantified under irradiation at various wavelengths using the FP-8300 spectrofluorometer from JASCO Corp, Tokyo, Japan. One sample of urine was irradiated with three wavelengths: 405 nm, 532 nm and 635 nm. These wavelengths were chosen to evaluate the urine’s fluorescent response across a wide range.

The frequency of the incident radiation is the parameter that most significantly impacts Raman intensity. Higher radiation frequencies result in a greater intensity of Raman scattered radiation from a sample. Since frequency is inversely proportional to wavelength, shorter wavelengths lead to an increase in Raman intensity [10,15]. That said, wavelengths larger than 635 nm were not evaluated.

The urine’s fluorescence response at each wavelength is presented in Figure 2, with the *x*-axis representing the wavelength range and the *y*-axis representing the fluorescence intensity.

As shown, urine exhibits a significant fluorescence when excited at 405 nm. Consequently, a laser with this wavelength is not suitable for Raman measurements due to extensive peak overlap and interference with the Raman signals. At 532 nm, the fluorescence intensity is lower, starting at 450 nm and rapidly decreasing after 532 nm. Despite the considerable fluorescence, Raman signals are stronger when radiation sources with shorter wavelengths are used. In contrast, excitation at 635 nm minimizes the fluorescence emitted but reduces the Raman signal intensity, making the spectral analysis more complex, especially while using low-power lasers.

Due to resource constraints, we were unable to investigate the impact of different wavelengths on the noise in Raman spectra. Nevertheless, the 532 nm wavelength was selected as the optimal choice, offering a balance between minimizing fluorescence interference and maximizing Raman signal intensity. Techniques such as spectral filtering and signal processing will be employed to further reduce fluorescence effects.

Several commercially available 532 nm lasers have narrow emission ranges and high power, but they can be highly expensive, exceeding the budget proposed for the low-cost system we intend to develop. Considering this analysis, the CPS532 compact low-power laser from Thorlabs was selected as the most cost-effective option for the Raman system. Its compact design and small size make it an ideal candidate for integration into portable systems. Operable within a temperature range of 10 °C to 40 °C, it has a typical power of 4.5 mW.

Due to heat dissipation during operation, the laser can reach high temperatures, potentially compromising its stability and spectral resolution. Considering this, a temperature control system was integrated into the laser to ensure consistent operating conditions, guaranteeing the laser’s stability, durability and spectra stability.

A custom support structure to accommodate the CPS532 laser and its control system was fabricated using 3D printing/milling processes. A Peltier element in the setup ensures the heat transfer between the laser module and a dissipator. A picture of the laser support is reported in Figure 3.

#### 2.1.2. Optical Elements

This system includes a series of filters and mirrors to achieve high-quality and high-resolution Raman spectra. All optical elements were purchased from Thorlabs.

The green radiation emitted by the laser begins its optical path by striking the directional mirror, model PF10-03-G01, which directs the beam towards the dichroic mirror. The dichroic mirror acts as a wavelength-selective filter, transmitting and reflecting light according to its wavelength. For this system, the dichroic mirror chosen was DMLP550, with a cutoff wavelength of 550 nm. This mirror reflects radiation below 550 nm, ensuring that the 532 nm laser radiation reaches the sample, and transmits all radiation above 550 nm. Both mirrors are mounted on KM100 kinematic brackets, allowing precise alignment of the laser beam with the sample and the spectrometer’s entrance.

Before reaching the sample, the laser beam passes through a lens integrated into the cuvette holder. This lens focuses the beam onto a small area of the cuvette, concentrating all the radiation on a specific region of the sample, and it also works as a collimator for the Raman radiation coming out from the sample.

Upon interacting with the sample, both Raman and non-Raman scattered radiation are emitted in all directions. A portion of this scattered radiation is captured by the lens of the cuvette holder and collimated towards the dichroic mirror. The Raman scattered radiation, with wavelengths in the yellow–red range (above 550 nm), passes through the dichroic mirror and is directed to the edge-pass filter.

The edge-pass filter, model FELH0550, removes any residual non-Raman radiation, with wavelengths below 550 nm, ensuring that only Raman radiation proceeds to the detector. The cutoff wavelength of the filter is precisely adjusted to extend the Raman spectrum while avoiding the excitation wavelength by slightly tilting it.

As the radiation passes through the dichroic mirror and edge-pass filter, it undergoes some horizontal displacement due to the thickness of these components. To correct this, the beam path is corrected using the WG11050-A compensation window.

After the realignment, the radiation passes through a set of lenses and a slit. The first achromatic lens, AC127-019-A, collects the Raman radiation focusing on an S50K slit with a 50 μm aperture, mounted in a CRM1T/M rotation cage. This slit ensures precise control of the amount of radiation that passes to the subsequent steps of the process, mainly defining the spectrometer resolution. A second achromatic lens, AC254-050-A, collimates the light exiting the slit and adjusts it to the grating surface.

The beam then reaches the GR25-1205 diffraction grating, with a density of 1200 lines per millimeter. This grating disperses the light into various directions based on its wavelength, allowing the analysis of the radiation spectrum.

Ultimately, the dispersed radiation is focused by an objective on the detector, which records the intensity at each position corresponding to specific wavelengths in the spectrum.

#### 2.1.3. Detector/Spectrometer

A FLIR Blackfly GigE camera is used as the detector, i.e., model BFLY-PGE-31S4M-C. This choice was based on its full compatibility with the selected data collection software, ensuring perfect integration of the camera into the developed system. This PointGrey model offers a resolution of 2048 × 1536 pixels with a CMOS-type sensor (Teledyne Vision Solutions, Richmond, BC, Canada). The focusing lens chosen to collect the Raman radiation was the 50 mm lens MVL50M23 from Thorlabs.

#### 2.1.4. Data Collection Software

The software chosen for Raman spectra acquisition is the Spectrum Analyzer, available on the OpenRAMAN website. This software was designed to operate with the type of system developed, minimizing risks of incompatibility and ensuring consistency in the spectra obtained. Additionally, the software is free and was previously validated for use with the selected camera model, guaranteeing compatibility with the system [16].

### 2.2. System Assembly

The assembly process is detailed on the OpenRAMAN website [13].

The baseplate was fabricated from an aluminum plate using computer numerical control (CNC) machining, according to the distances and dimensions specified on the OpenRAMAN platform. The cuvette holder used in the system is the CVH100/M model, chosen for its versatility, allowing the integration of various analysis techniques to examine the same sample.

To protect the Raman system instrumentation from external interference, two custom-made covers were designed to enclose the entire optical system. These covers were manufactured using a 3D printer, ensuring effective protection against light and external impurities. Below, Figure 4 presents an image of the top view of the developed Raman system without the covers. The alignment and calibration of the Raman system were performed following the OpenRAMAN guidelines.

### 2.3. Optimization of System Components

#### 2.3.1. Laser Operating Temperature Optimization

The implementation of the temperature control system also aimed to optimize the emission spectrum of the CPS532 laser. Adjusting the laser’s operating temperature can enhance its spectral resolution and narrow its emission range, making it more efficient for Raman analysis.

To evaluate the laser’s behavior at various temperatures, two experiments were conducted. The first focused on understanding the effect of temperature on the full width at half maximum (FWHM) of the laser peak. The laser spectrum was acquired using the CCS200/M spectrometer and the SMA905 fiber optic cable, which captured and transmitted the laser beam to the spectrometer. The fiber was integrated into the CVH100/M cuvette holder. The spectrometer has its own software, ThorSpectra. After setting up the equipment and connecting the spectrometer to the software, the laser spectrum was acquired at operating temperatures of 20 °C, 25 °C, 30 °C, 35 °C and 40 °C. To prevent signal saturation, all spectra were acquired using an integration time of 0.1 ms.

At an operating temperature of 20 °C (cf. Figure 5), the CPS532 laser spectrum exhibits two peaks: a primary peak at 532.8 nm, with an intensity of 0.90 and an FWHM of 1013.7 pm, and a secondary peak at 537.3 nm, with an intensity of 0.04. The secondary peak is attributed to the use of the SMA905 optical fiber, which, despite introducing this interference, is essential for ensuring the reproducibility and comparability of the results. Notably, the optical fiber will not be used for Raman spectra acquisition, so this interference will not affect the quality or accuracy of the spectra obtained with the system.

Subsequently, the laser’s operating temperature was varied to 25 °C, 30 °C, 35 °C and 40 °C, with the obtained spectra presented in Appendix A. Figure 6 illustrates the variation in the FWHM as a function of the laser’s operating temperature.

Contrary to expectations, the relationship between the CPS532 laser’s operating temperature and its FWHM is non-linear. The FWHM decreases until 30 °C and increases at higher temperatures, peaking at 40 °C, the laser’s maximum operating temperature. At 40 °C, the main peak exhibits instability in shape and a significant intensity reduction, dropping to 0.51. These results indicate that high temperatures negatively affect the laser performance, making operation under such conditions undesirable. At 30 °C, the FWHM of the main peak reaches its minimum value of 649.8 pm, with an acceptable intensity of 0.82. Thus, 30 °C is the temperature that minimizes the spectral range and that, consequently, increases the spectral resolution of Raman spectra.

In a second phase, the spectrum of ethanol (chemical formula: CH_3_CH_2_OH) was acquired using the developed Raman system at three different laser operating temperatures: 25 °C, 30 °C and 35 °C (cf. Figure 7). The main objective of this phase was to assess how the laser’s operating temperature influenced the quality and accuracy of Raman spectra. The ethanol spectrum was acquired with Spectrum Analyzer software 1.2, with the acquisition parameters reported in Table 1. The visualization of the spectrum obtained was performed using SpectraGryph software, version 1.2.

The presence of characteristic ethanol peaks in the spectrum, although not highly intense, demonstrates that the system operates with sufficient precision to capture molecular vibration signals from the sample.

When comparing the three ethanol spectra, the spectrum obtained at 35 °C shows the lowest peak intensities, particularly in zone (4), around 2800 to 3000 cm^−1^. Between the spectra obtained at 25 °C and 30 °C, the peaks at 25 °C exhibit slightly lower intensity. Considering that the first optimization phase identified 30 °C as the optimal operating temperature for enhancing laser performance and given the quality of the ethanol spectrum acquired at this temperature, we concluded that 30 °C maximizes the overall efficiency of the developed Raman system.

#### 2.3.2. Optimization of Spectrum Analyzer Software Acquisition Parameters

The software used allows for adjustments of acquisition settings, such as exposure time and the average number of images. These adjustments are critical for ensuring result stability and for obtaining high-quality spectra with a strong signal-to-noise ratio (SNR).

For this optimization, the ethanol Raman spectrum was acquired using different acquisition parameters, with the laser operating at 30 °C. Each parameter was varied three times, as reported in Table 2.

In the first stage, the exposure time, which controls the time during which the PointGrey camera “captures” light, was evaluated. It is important to note that the maximum exposure time allowed by the software is 31.6 s. Additionally, spectra acquired with exposure times shorter than 2.1 s exhibited high noise levels, which made their analysis exhaustive and confusing.

The shortest exposure time of 2.1 s resulted in a Raman spectrum with high noise, which interfered with the ethanol peaks by artificially increasing their intensity. However, this increase was solely due to the added noise, compromising the spectrum analysis. The noise may have originated from the instrumentation, including laser or thermal fluctuations or electronic noise. Increasing the exposure time to 10 s significantly reduced the noise, as expected, since a longer acquisition time allowed the camera to integrate the incoming light, thus increasing the signal-to-noise ratio. Further increasing the acquisition time to the maximum value, 31.6 s, caused subtle differences in noise levels, especially in the range between 1500 cm^−1^ and 2800 cm^−1^. Although this longer exposure did not substantially improve the ethanol spectrum, it can be beneficial for urine spectra, which tend to have a high noise level. Based on this, we conclude that an exposure time of 31.6 s is optimal for improving Raman spectra quality, especially for urine analysis. See Figure 8.

Then, the impact of the number of images acquired and combined to generate the final spectrum was evaluated; the parameters are reported in Table 3 and the obtained spectra are reported in Figure 9. The analysis of the obtained spectra reveals that when only two images are acquired, the noise is significantly higher, particularly in regions without peaks, complicating the analysis of more complex spectra, such as urine spectra. Increasing the number of images to 12, the noise level in these regions decreases considerably while preserving the intensity of ethanol’s characteristic peaks. This provides the highest SNR ratio, as it minimizes noise while maintaining distinct peaks. Acquiring spectra with a higher number of images resulted in a similar noise level in the regions without peaks. However, the intensity of the peaks reduced considerably. Therefore, we conclude that the average number of images that improve the overall quality of the spectrum while maintaining a high SNR ratio is 12 images.

The gain parameter, which represents the amplification of the electrical signal generated by the camera’s sensor after capturing the light, was also evaluated. However, the ethanol spectra did not exhibit significant variations in noise or intensity, making it impossible to draw concrete conclusions about its impact.

This study faced limitations, as it was not feasible to explore all possible parameter values. However, the optimal settings to enhance the quality of the ethanol spectrum were 31.6 s as the exposure time and an average of 12 images.

### 2.4. Role of the Laser’s Power in the Raman Spectrum

Maintaining the overall hardware configuration unchanged, we tested our system with a new laser with a higher power, exceeding 80 mW, and a short spectral linewidth (nominal FWHM = 0.003 nm), which is manufactured by Frankfurt Laser Company, i.e., model FPYL-532-80T-LN-S. To evaluate its performance compared to the previously used laser, we acquired the ethanol spectrum under similar conditions (cf. Figure 10).

The differences between the spectra are notable, with the higher-powered laser providing a substantial enhancement in the intensity of the Raman peaks, lower noise and shorter acquisition times. Using the CPS-532 laser, the maximum intensity achieved was only 0.035, which was notably lower. In contrast, the FPYL-532 laser allows for intensities to reach up to 4, representing a remarkable improvement. This enhancement directly improves the quality and clarity of the obtained data, highlighting the critical role of laser power in Raman spectroscopy analyses.

By analyzing the acquired spectra, we may conclude that the resolution of the system with CPS-532 is about 15 cm^−1^; a similar value was obtained with FPYL-532, being the main advantage of the higher signal/noise ratio. Further improvements in the resolution could be obtained by taking advantage of the enhanced laser power for reducing the slit size. However, the FPYL laser is significantly more expensive than the one employed in this system, making it unsuitable for low-cost approaches, and it has been used only as a comparison term. Once the laser operation entered a constant temperature condition, no significant fluctuations in the peak positions were observed, which can be considered lower than the system resolution.

The spectral resolution of available commercial Raman portable setups ranges from 8 cm^−1^ for the Wasatch WP532X (532 nm) to 9.7 cm^−1^ for the Thorlabs Portable Coded-Aperture Raman Spectrometer RASP2 (682 nm) and 10 cm^−1^ for the Hamamatsu C15471 (785 nm). All these systems are extremely compact and efficient, but they are available at a price level that is too high to address the large-scale employment of a POC system.

While a research-level Raman spectrometer can present an impressive spectral resolution at less than 1 cm^−1^ over a wide spectral range (see, for example, the Horiba LabRAM Soleil), when compared with top-class Raman equipment, one major advantage of the simplicity of the mechanical parts are the soft requirements for precise alignment, which support the stability we observed over time and the low sensitivity to environmental conditions, like humidity and temperature. Targeting our application playground, once the spectral range of interest is defined within a specific clinical context, as assessed by a large-scale clinical study, the spectral resolution can be improved by tailoring the properties of the diffraction grating, sacrificing the spectral range.

## 3. Urine Spectrum Acquisition

To validate the developed system for urine spectra acquisition, its performance was evaluated in this context. The focus of this evaluation was not to identify the biomarkers present in the urine, nor to conduct their detailed analysis, but to ensure that the Raman system is able to acquire urine spectra with precision and consistency. For this purpose, five samples of urine were used. The acquisition parameters used are described in Table 4, and the spectra are reported in Figure 11.

By analyzing the obtained spectra, it is evident that all five samples exhibit consistent peaks around 415 cm^−1^ and 735 cm^−1^. Regarding the 415 cm^−1^ peak, a limited amount of information is available in the literature; however, this peak is likely associated with the presence of glycogen in urine, which typically exhibits a Raman peak around 480 cm^−1^ [17]. The 735 cm^−1^ peak is probably related to creatinine, despite its usually reported values being around 640 cm^−1^ and 700 cm^−1^ [9,18,19,20]. Additionally, a broader and less intense peak is observed between 990 cm^−1^ and 1090 cm^−1^, likely associated with urea, which typically presents Raman peaks around 1000 cm^−1^ and 1006 cm^−1^ [9,18,19]. The intensities of these peaks varied across the samples, demonstrating the system’s sensitivity to the differing compound concentrations in urine and its consistency in the acquisition of the spectra. Overall, the intensity of the peaks remained relatively low, with noise present throughout the spectra. Enhancing peak intensity and reducing noise would require a laser with higher spectral resolution and greater power.

This analysis demonstrates that the developed Raman system is both consistent in its spectra acquisitions and sensitive to variations in compound concentrations within a complex matrix. These results reinforce the system’s potential to detect changes in urine composition, essential for its application in future diagnostic analyses.

## 4. Data Treatment and Validation

To validate that the system runs a correct spectra acquisition process, a typical method is to compare the results with the production data of a well-known composition. So, following this well-established procedure, we tested our Raman system by comparing its results with other data published in the literature about simple alcohols, like ethanol and methanol. The Raman analysis of these alcohols was largely developed to address the problem of detecting the so-called “fake wine” production, which is often accompanied by some methanol components, producing toxicity for excessive dosage [21]. The importance of the clinical problem associated with wine adulteration justifies the abundance of Raman data available. We may conclude that the results obtained for both alcohols agree with the data available in the literature [22], as confirmed below in Figure 12. It is possible to operate a Gaussian convolution on the raw data to obtain a clean plot of the spectra, as reported in Figure 13. While the readability of the plots is greatly improved, the optimal filtering parameters are dependent on the data quality and the number and intensity of the peaks, so they can be inferred correctly only in a second phase; otherwise, the blurring operation directed to remove noise may become associated with the risks of removing relevant details. This is evident in the treatment of the urine samples, which are much richer in content components and, correspondingly, in the number of peaks, as depicted in Figure 14. If the complete composition of a urine sample is not known a priori, the filtering operation should be executed only within a clinical framework, having previously defined which peaks need to be searched. To obtain more information on this topic, a clinical study, with urine samples collected in a hospital emergency department, is foreseen for the future. If enough urine samples are available with an associated clinical diagnosis, it will be possible to conduct a direct search of the clinically significant peaks. Within this framework, instead of using Raman spectroscopy for a detailed chemical description of the urine contents, it will be possible to establish a correlation between a specific disease risk and the presence and intensity of determinate peaks. This consideration drives the attention directly to the development of a classification scheme suitable to be elaborated by a machine learning model. To support this point of view and demonstrate the feasibility of a machine learning approach for Raman spectra interpretation, in the following section, we report a case study elaborating on the detection of the methanol percentage in an ethanol solution.

The analysis of urine samples is interesting as a proof of concept, pointing the attention to an application domain where a portable Raman analysis can contribute to addressing an important social problem. However, the peak intensities and distributions in the spectra of urine samples can present a large variability depending on the specific condition of the donors, so they cannot be used to objectively infer the performance level of the prototype. So, as a comparison with other data reported in the literature for methanol detection using portable Raman systems, Table 5 reports the level of detection (LOD) and some technical specifications of the systems. Both the CPS532 and FPYL532 versions are included in the table. As a reference of values, it is worth recalling that drinks made from “industrial methylated spirits” [5% (*v*/*v*) methanol:95% (*v*/*v*) ethanol] can cause severe and even fatal illness, and the current EU general limit for naturally occurring methanol is 10 g methanol/L ethanol (which equates to 0.4% (*v*/*v*) methanol at 40% alcohol) [23]. Regarding the low-power version with the CPS532 laser, the LOD is higher than the EU recommendation, as the 2840 cm^−1^ peak remains hidden by the noise for a concentration lower than 4.76% of methanol. Moving to the higher-power FPYL532 (leaving all the other components unchanged), the S/N is much improved, and the LOD falls to a value below 0.25%. It should be noted that the 80 mW of laser power is still well below the standard hundreds of mW normally used in infrared models, making the management of the safety requirements for a portable system easier.

## 5. Development of the Supervised Learning Model

### 5.1. Data Acquisition

By developing a learning model, we aim to evaluate its capability to differentiate and classify Raman spectra. If successful, the model could contribute to making the diagnosis of kidney diseases simpler, faster and more effective.

To train the model, Raman spectra of six methanol–ethanol solutions with different compositions were acquired. The decision to use methanol–ethanol solutions was based on the fact that they generate simpler Raman spectra. This characteristic facilitated the initial development and optimization of the algorithm, as it allowed us to evaluate the model’s ability to classify straightforward Raman signals before progressing, in the future, to more complex biological samples.

The methanol–ethanol spectra were acquired using the developed Raman system. To ensure dataset diversity, the acquisition parameters (exposure time, gain and ROI) of the Spectrum Analyzer software were varied between each acquisition. A total of 618 spectra (cf. Table 6) were acquired and divided into two subsets: 80% (495 spectra) for model training and 20% (123 spectra) for model testing. Each spectrum was converted into a 2D image with a resolution of 1000 × 2000 pixels, enabling the model to extract spectral features for correct classification.

Next, we present a graph showcasing an example spectrum for each alcohol solution used in training the supervised learning model. This visualization highlights the spectral differences among the six solutions, exemplifying the variability in the dataset used. The acquisition parameters used for the shown alcohol spectra are listed in Table 7, and the data are reported in Figure 15.

### 5.2. Supervised Learning Model Development

The learning model was developed using MATLAB Version 2023, and it is a Convolutional Neural Network (CNN) tailored to classify Raman spectra.

As previously mentioned, the input data consists of 2D images of the Raman spectra from alcohol solutions, all standardized to the same dimensions and represented with three color channels (RGB). The network architecture, as depicted in Figure 16, includes two hidden layers following the input layer. Each hidden layer comprises a convolutional layer, a batch normalization layer, a ReLu activation function and a pooling layer. The output layers, essential for the final spectra classification, consist of a fully connected layer, followed by a SoftMax activation function and a classification layer.

### 5.3. Supervised Learning Model Optimization

Hyperparameters significantly influence a model’s performance, particularly its training speed and accuracy. These hyperparameters can optimize critical aspects of a model, making it more robust and efficient.

The MaxEpoch hyperparameter defines the maximum number of training iterations, i.e., epochs. Higher epoch values allow a model to learn more effectively but increase the risk of overfitting and extend the training time. During an epoch, a model processes the entire training set, dividing it into mini-batches, another key hyperparameter. The Mini-batch size impacts training efficiency, as larger groups can result in a longer training time.

The InitialLearnRate hyperparameter defines the rate at which a model adjusts its weights during training. This rate directly influences the speed and effectiveness of the training process. An improperly adjusted rate can prevent a model from converging and learning from the training data.

The model was optimized by varying the mentioned hyperparameters, and their impact was evaluated based on the training time, accuracy and precision of the model (cf. Table 8). Despite its simplicity, this empirical approach allows for the identification of the most effective hyperparameters for our dataset and model architecture.

Analyzing the results, the hyperparameter with the greatest influence on the model’s accuracy and precision is InitialLearningRate. When this parameter was increased from 0.001 to 0.01, the model’s performance declined. With a rate of 0.1, its accuracy dropped significantly to 17.07% and precision to 16.67%, as excessively high learning rates can cause large weight adjustments, preventing the model from adequately capturing data patterns. Thus, the optimal learning rate for the developed model is 0.001.

Adjusting the Mini-batch size especially influenced the training time. With 16 batches, the model achieved 100% accuracy and precision, within approximately 5 min of training. These values reflect the limitations imposed by the small dataset size used, which likely contributed to overfitting.

The MaxEpoch hyperparameter did not behave as anticipated. Increasing the number of epochs from 20 to 30 resulted in a slight decrease in performance, indicating that the model could be overfitting to the training data. On the other hand, with the decrease in epoch cycles from 20 to 10, the model’s accuracy and precision stayed equal, though it minimized training time to just about 2 min.

In summary, the hyperparameter combination that maximizes model performance, while maintaining efficient training times and avoiding overfitting, is MaxEpoch of 10, InitialLearningRate of 0.001, and MiniBatchSize of 16.

These results highlight the potential of the developed learning model for Raman spectrum classification, evidencing its capability to identify patterns in these spectra.

## 6. Conclusions and Future Perspectives

This study demonstrated the potential of a simplified and portable Raman spectroscopy system for acquiring the spectra of liquid and complex samples, including urine. The application of this system could simplify and enhance the accessibility of diagnosing diseases outside clinical contexts. Future work will focus on the identification and analysis of specific biomarkers associated with kidney diseases using the developed system.

The system’s main challenge was the high noise levels and the low intensity of the characteristic peaks in the spectrum obtained. Overcoming this issue would require a higher-power laser since the one used in this study had a relatively low power of 4.5 mW. Additionally, urine’s fluorescence spectrum indicates a low fluorescence intensity when irradiated with a red laser with 635 nm, suggesting the potential development of a Raman system using a high-power laser with this emission wavelength.

The assembly, alignment and calibration of the Raman system were successfully achieved at this stage. In a future phase, the use of more sophisticated optical elements could be explored to amplify Raman scattering. Similarly, the acquisition parameters were optimized using ethanol spectra; however, their direct impact on urine spectra should be explored in subsequent phases.

During the optimization phases, the system exhibited high stability, as evidenced by minimal spectral variations across repeated measurements over time. Despite higher noise levels in some instances, the system successfully acquired the ethanol spectrum with its characteristic Raman peaks, demonstrating its robustness and reliability under challenging conditions.

The analysis of the five urine spectra with the developed system demonstrates consistency and sensibility to variations in compound concentrations in urine. Future work will involve expanding the dataset to include urine samples from healthy and unhealthy individuals, as well as validating the system’s performance in detecting clinically relevant biomarkers.

The implementation of a supervised learning model for spectra classification proved to be an effective tool, capable of recognizing patterns in Raman spectra and achieving accurate classification. The application of these kinds of models can facilitate the differentiation between healthy and pathological spectra. The results obtained with methanol–ethanol solutions demonstrated the model’s ability to classify Raman spectra. These results serve as a foundation for future work, where the model will be trained with urine spectra to identify kidney disease biomarkers.

Although a direct comparison with commercial Raman instruments was not performed in this study, the developed system is significantly more affordable than the systems available on the market, having an estimated cost below EUR 5000. Furthermore, its design allows for portability, highlighting its potential for point-of-care applications.

Despite the limitations imposed by instrumentation and budget, this work paves the way for the development of a Raman spectroscopy system that, combined with machine learning, creates an accessible, portable and effective tool for non-invasive diagnosis, particularly for kidney disease detection.

## Figures and Tables

**Figure 1 sensors-25-00659-f001:**
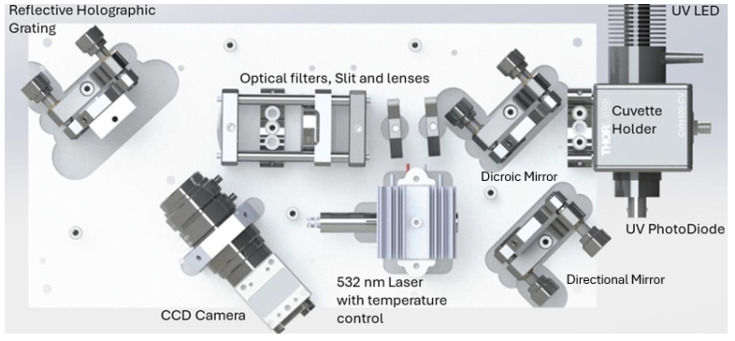
Overview of the developed system, adapted from [14]. (Copyright 2025 Society of Photo-Optical Instrumentation Engineers (SPIE). One print or electronic copy may be made for personal use only. Systematic reproduction and distribution, duplication of any material in this publication for a fee or for commercial purposes, and modification of the contents of the publication are prohibited).

**Figure 2 sensors-25-00659-f002:**
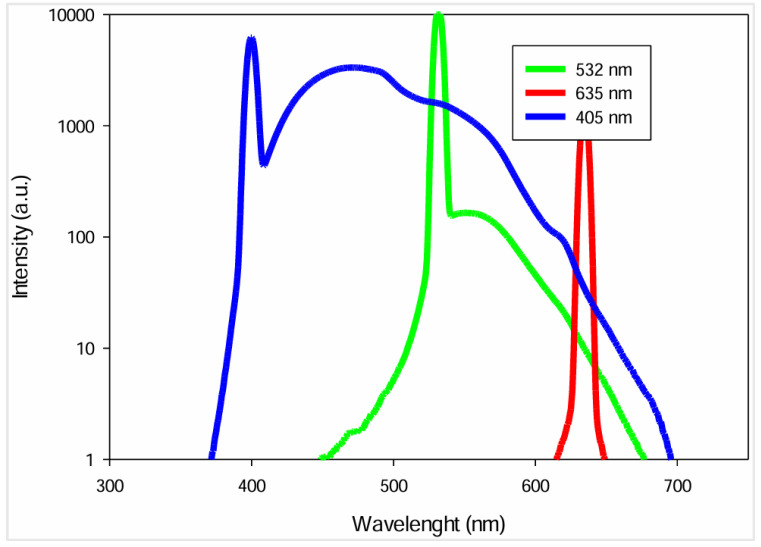
Urine fluorescence spectra at 3 excitation wavelengths [14]. (Copyright 2025 Society of Photo-Optical Instrumentation Engineers (SPIE). One print or electronic copy may be made for personal use only. Systematic reproduction and distribution, duplication of any material in this publication for a fee or for commercial purposes, and modification of the contents of the publication are prohibited).

**Figure 3 sensors-25-00659-f003:**
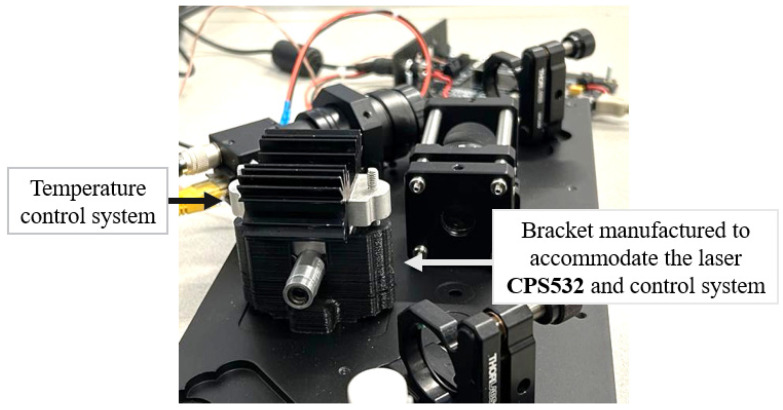
Detail of the support manufactured to accommodate the CPS532 laser and the temperature control system.

**Figure 4 sensors-25-00659-f004:**
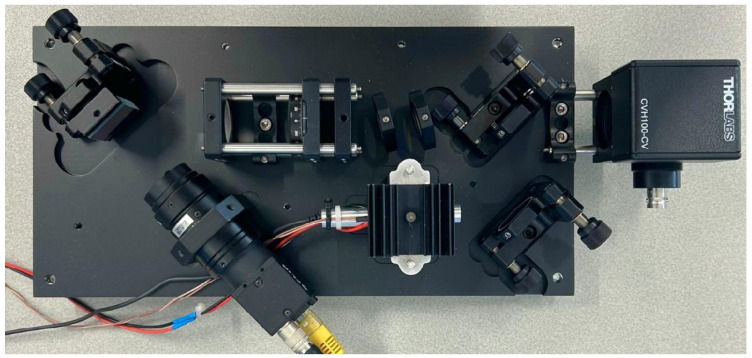
Top view of the developed Raman system.

**Figure 5 sensors-25-00659-f005:**
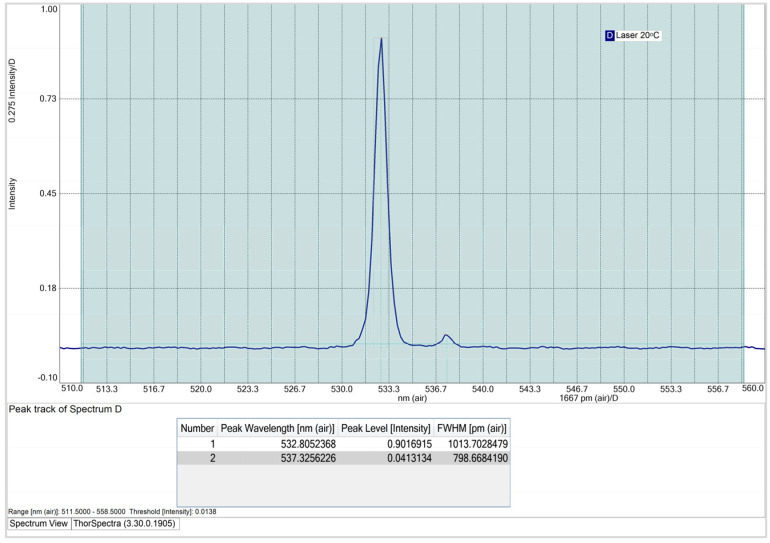
CPS532 laser spectrum at 20 °C, using the “Peak Track” tool from ThorSpectra.

**Figure 6 sensors-25-00659-f006:**
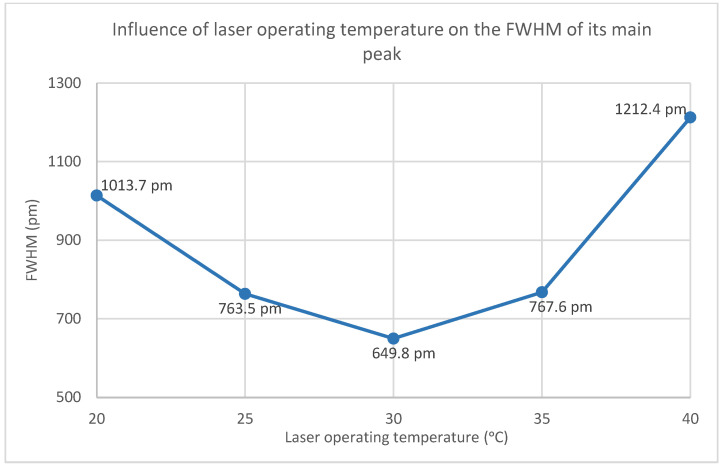
Influence of laser operating temperature on the FWHM of its main peak.

**Figure 7 sensors-25-00659-f007:**
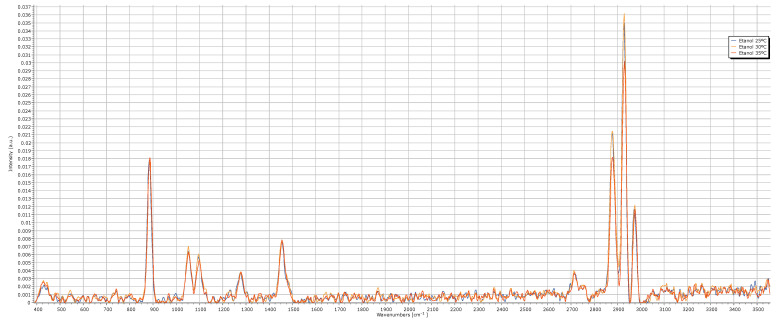
Raman spectrum of ethanol at laser operating temperatures of 25 °C, 30 °C and 35 °C.

**Figure 8 sensors-25-00659-f008:**
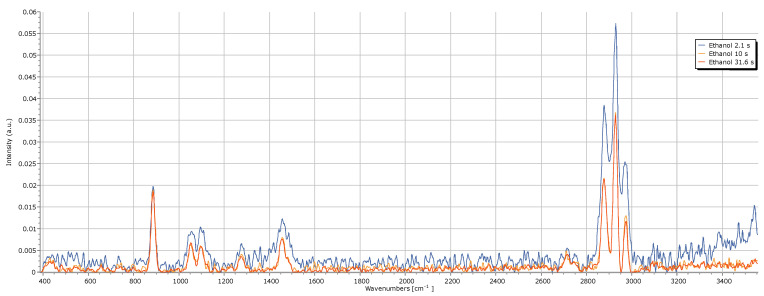
Impact of exposure time on the quality of the ethanol Raman spectrum.

**Figure 9 sensors-25-00659-f009:**
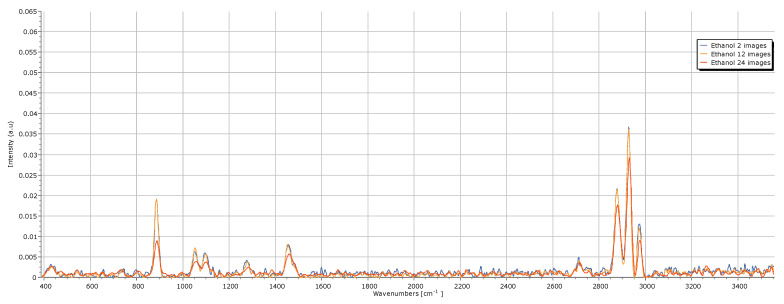
Impact of the average number of images on the quality of the ethanol Raman spectrum.

**Figure 10 sensors-25-00659-f010:**
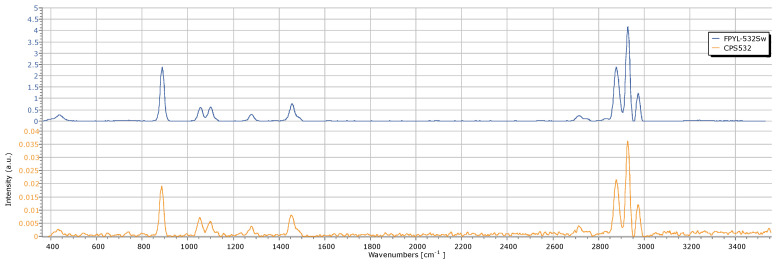
Comparison of the ethanol spectrum obtained with two different lasers.

**Figure 11 sensors-25-00659-f011:**
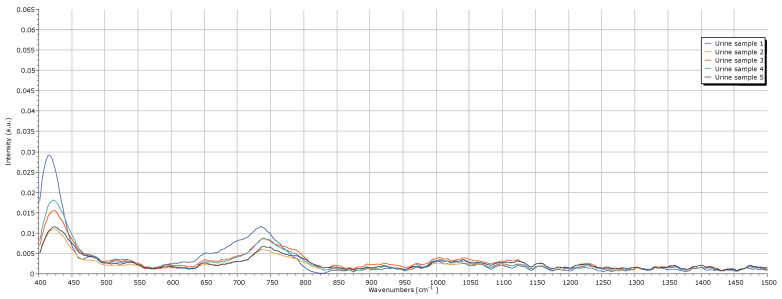
Raman spectrum of the five urine samples.

**Figure 12 sensors-25-00659-f012:**
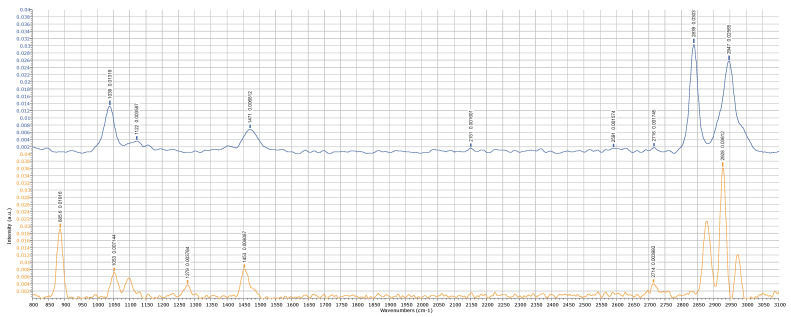
Raman spectrum of methanol (blue line) and ethanol (orange line). The relevant peaks, as described in the literature, are highlighted.

**Figure 13 sensors-25-00659-f013:**
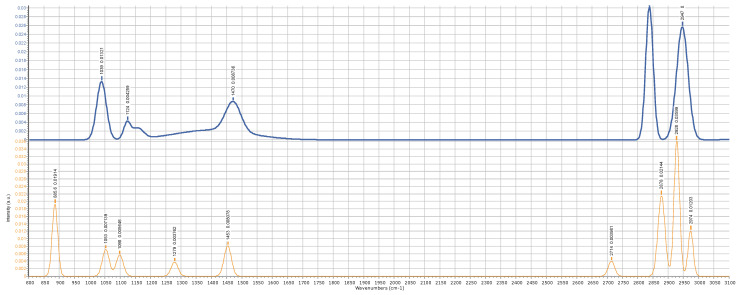
Raman spectrum of methanol (blue line) and ethanol (orange line). The data are filtered with a Gaussian convolution algorithm for noise removal.

**Figure 14 sensors-25-00659-f014:**
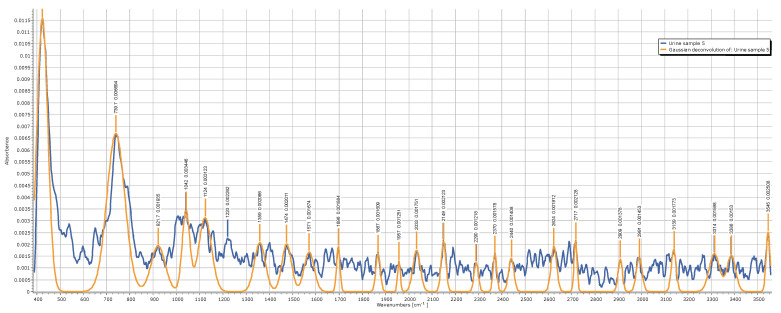
Raman spectrum of one urine sample and its corresponding filtered plot, obtained by a Gaussian convolution algorithm.

**Figure 15 sensors-25-00659-f015:**
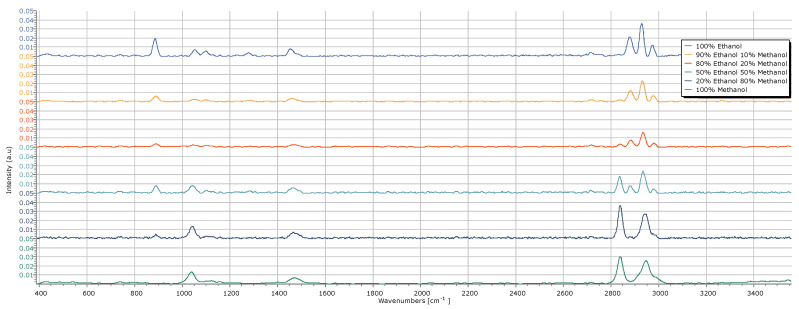
Example Raman spectra of the alcohol solutions used.

**Figure 16 sensors-25-00659-f016:**
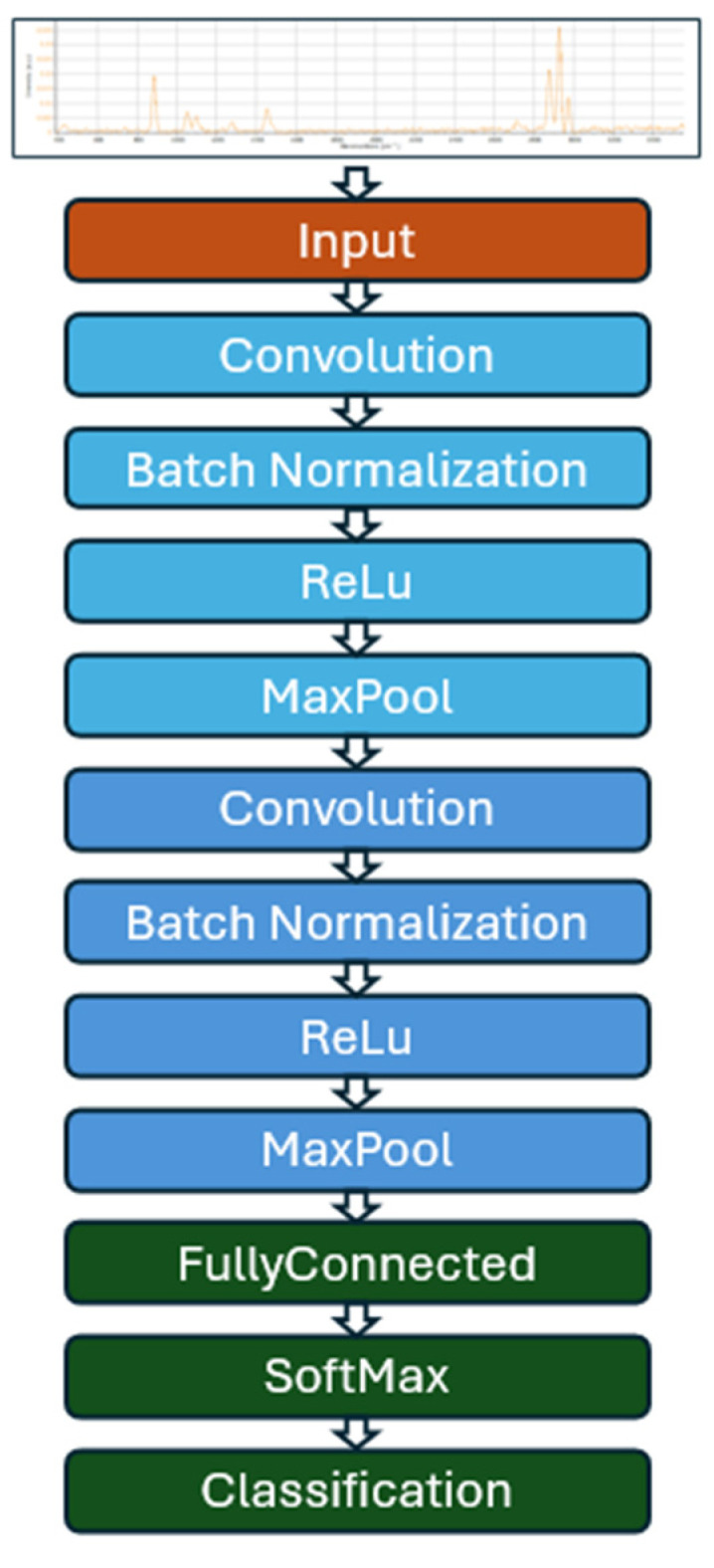
Supervised learning model architecture.

**Table 1 sensors-25-00659-t001:** Acquisition parameters used for the acquisition of the ethanol spectrum.

Exposure Time(s)	Gain (dB)	ROI (px)	Average Number of Images
31.6	11.5	128	3

**Table 2 sensors-25-00659-t002:** Acquisition parameters used to study the impact of exposure time on the quality of the ethanol Raman spectrum.

Name	Exposure Time(s)	Gain (dB)	Average Number of Images
Ethanol 2.1 s	2.1	3.8	2
Ethanol 10 s	10	3.8	2
Ethanol 31.6 s	31.6	3.8	2

**Table 3 sensors-25-00659-t003:** Acquisition parameters used to study the impact of the average number of images on the quality of the ethanol spectrum.

Name	Exposure Time(s)	Gain (dB)	Average Number of Images
Ethanol 2 images	10	3.8	2
Ethanol 12 images	10	3.8	12
Ethanol 24 images	10	3.8	24

**Table 4 sensors-25-00659-t004:** Acquisition parameters used for the acquisition of urine spectra.

Exposure Time(s)	Gain (dB)	ROI (px)	Average Number of Images
31.6	1.5	128	10

**Table 5 sensors-25-00659-t005:** Level of detection (LOD) of methanol in a solution with ethanol.

Model	LOD (%)	LOD (g/L)	Laser Wavelength (nm)	Laser Power (mW)	More Significant Peak (cm^−1^)
CPS532	4.76	37.67	532	5	2840
FPYL532	0.25	1.97	532	80	2840
Ref. [24]	0.23–0.39		830	400	1030
Ref. [25]	0.025		1064	450	1023
Ref. [22]	20		532	40	2840

**Table 6 sensors-25-00659-t006:** Data used for the supervised learning model.

Solutions	Number of Spectra Obtained	Total No. of Data
100% Ethanol	106	618
90% Ethanol 10% Methanol	103	
80% Ethanol 20% Methanol	103	
50% Ethanol 50% Methanol	102	
20% Ethanol 80% Methanol	102	
100% Methanol	102	

**Table 7 sensors-25-00659-t007:** Acquisition parameters used for the acquisition of the example alcohol spectra.

Exposure Time(s)	Gain (dB)	ROI (px)	Average Number of Images
10	3.8	128	12

**Table 8 sensors-25-00659-t008:** Analysis of how changing hyperparameters impacts model evaluation.

Hyperparameters of Training			Model Performance		
MaxEpoch	InitialLearningRate	MiniBatchSize	Accuracy (%)	Precision (%)	Training Time
20	0.001	64	96.75	96.79	24 min 04 s
20	0.001	32	98.37	98.71	10 min 52 s
20	0.001	16	100	100	05 min 31 s
30	0.001	16	99.19	99.21	08 min 20 s
10	0.001	16	99.19	99.21	02 min 45 s
20	0.1	16	17.07	16.67	05 min 28 s
20	0.01	16	87.81	87.78	05 min 30 s

## Data Availability

The data used in this project as well as a version of the algorithm are available upon request.

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
