# Peer review of "Low-Cost Raman Spectroscopy Setup Combined with a Machine Learning Model"

_sensors, 2025, doi:10.3390/s25030659_

Round 1
Reviewer 1 Report
Comments and Suggestions for Authors
The manuscript has significant potential, but it requires additional work to provide more experimental data, detailed comparisons with existing technologies, and a clearer description of the machine learning model. The focus on kidney disease diagnostics should be supported by relevant data or clearly positioned as a future direction. Moreover, figures, language, and methodology details need refinement to enhance clarity and scientific precision.
1. The manuscript lacks clarity and detailed explanations regarding the tested ethanol and urine samples. It is unclear whether the presented Raman spectra represent an average of multiple samples or a single measurement (check fig 12). A more comprehensive discussion is required to describe the experimental methodology, including how many samples were tested, whether the spectra were averaged to account for variability, and how representative the results are of the broader dataset. This information is crucial for ensuring reproducibility and validating the reliability of the findings.
2. Which urine samples has been tested they are taken from normal or sick person (not clear)?
3. The manuscript briefly mentions the use of a 532 nm laser but does not analyze how noise in Raman spectra impacts the results.
4. The description of the machine learning model is insufficient. Key details such as the dataset size, diversity, training-validation split, and hyperparameter tuning are not provided.
5. Authors stated that”This work is part of a project, whose future goal is the development of a portable optoelectronic system to determine the patient's renal status. This system will have three approaches to urine analysis, the present article focuses exclusively on the Raman spectroscopy part’’, no need to write this sentence in the manuscript.
6. The manuscript focuses on kidney-related applications (in the abstract), but there is no specific data or biomarkers related to kidney diseases presented in the study.
7. The study's focus on kidney patients is valuable; however, the proposed Raman spectroscopy setup and machine learning model have broader potential for diagnosing various diseases. Expanding the discussion to include this versatility would enhance the manuscript’s impact and demonstrate the wider clinical applicability of the system.
8. The machine learning model description and its training dataset seem underexplored and require further clarification.
9. Consider adding a numerical summary of the system’s performance metrics in abstract.
10. The authors should elaborate on the customization process for specific components.
11. Compare your findings with those from similar studies, emphasizing the novelty and improvements of the proposed system.
12. Improve the graphical quality and ensure consistent formatting throughout the text.
13. Please ensure uniform formatting of references
Suggestion: I suggest that the authors focus primarily on the Raman spectroscopy setup and the optimization of its parameters in this manuscript. While the application is mentioned, the lack of detailed information and supporting data makes it less convincing. A thorough explanation of the setup and its optimized performance would provide a stronger foundation for future discussions on its practical applications.
Comments on the Quality of English Languageshould be improved
Author Response
We thank the reviewer for the detailed review out for the manuscript. In this document, we report our answers to each comment, comments indicating the changes in the manuscript, also highlighted in yellow in the nmansucript. We hope to have improved the paper and made it more robust, comprehensive and easier to understand by addressing these comments.
- The manuscript lacks clarity and detailed explanations regarding the tested ethanol and urine samples. It is unclear whether the presented Raman spectra represent an average of multiple samples or a single measurement (check fig 12). A more comprehensive discussion is required to describe the experimental methodology, including how many samples were tested, whether the spectra were averaged to account for variability, and how representative the results are of the broader dataset. This information is crucial for ensuring reproducibility and validating the reliability of the findings.
Figure 12 shows an example of one spectrum for each alcohol solution used for the development of the model. We listed the acquisition parameters in Table 6, Section 4.1.1, lines 418 – 422. The missing information was added according to suggested.
- Which urine samples has been tested they are taken from normal or sick person (not clear)?
The urine samples used in this study were obtained from patients diagnosed with kidney diseases. However, the clinical analysis of the urine spectra is outside the goal of the present article. A follow-up study over a larger set of urine samples collected at the emergency department of a public hospital is planned to start in 2025.
- The manuscript briefly mentions the use of a 532 nm laser but does not analyze how noise in Raman spectra impacts the results.
The laser’s impact on Raman spectra noise is an important study, however, due to resource limitations, we were unable to analyze the effect of different wavelengths on the spectra noise. The 532 nm laser was chosen based on its balance between the fluorescence emission of urine and the Raman scattering intensity, as predicted by the Raman intensity equation. We added this information in Section 2.1.1, lines 144 – 146.
- The description of the machine learning model is insufficient. Key details such as the dataset size, diversity, training-validation split, and hyperparameter tuning are not provided.
We included more details about the development of the machine learning model, such as dataset set, diversity and training-validation split (Section 4.1.1, lines 408 – 414). The hyperparameter tuning was also better explained in Section 4.1.3, lines 452 – 455.
- Authors stated that ”This work is part of a project, whose future goal is the development of a portable optoelectronic system to determine the patient's renal status. This system will have three approaches to urine analysis, the present article focuses exclusively on the Raman spectroscopy part’’, no need to write this sentence in the manuscript.
As suggested, we removed this sentence from Section 2.
- The manuscript focuses on kidney-related applications (in the abstract), but there is no specific data or biomarkers related to kidney diseases presented in the study.
While the future goal of this study is to support kidney disease diagnostics, the current article focuses primarily on the development, optimizations and validation of a low-cost Raman spectroscopy system and a machine learning model. This information was included in Section 5, lines 481 – 483 and lines 501 – 504.
- The study's focus on kidney patients is valuable; however, the proposed Raman spectroscopy setup and machine learning model have broader potential for diagnosing various diseases. Expanding the discussion to include this versatility would enhance the manuscript’s impact and demonstrate the wider clinical applicability of the system.
We agree that the proposed Raman spectroscopy system and supervised learning model have broader potential for diagnosing various diseases, we have elaborated this question in Section 1, lines 96 – 100.
- The machine learning model description and its training dataset seem underexplored and require further clarification.
The architecture of the supervised learning model was added in the new Section 4.1.2, lines 430 – 438.
- Consider adding a numerical summary of the system’s performance metrics in abstract.
As suggested, we added the performance metrics of the model in Abstract, lines 29-31.
- The authors should elaborate on the customization process for specific components.
We have elaborated the customization process, the laser support was fabricated with 3D printing/milling processes (Section 2.1.1 line 161), the baseplate was fabricated from an aluminum plate, using CNC (Section 2.2, lines 221 – 222) and the covers were manufactured using 3D printing (Section 2.2, lines 227 – 228).
- Compare your findings with those from similar studies, emphasizing the novelty and improvements of the proposed system.
In this study, we didn’t perform a comparison between our system and commercial Raman instruments, however we have added a paragraph explaining the most significant difference, its affordability, in Section 5, lines 511-514.
- Improve the graphical quality and ensure consistent formatting throughout the text.
The graphical quality and consistency throughout the text were improved, thank you.
- Please ensure uniform formatting of references
We have revised the references to ensure consistent formatting, thank you.
Suggestion: I suggest that the authors focus primarily on the Raman spectroscopy setup and the optimization of its parameters in this manuscript. While the application is mentioned, the lack of detailed information and supporting data makes it less convincing. A thorough explanation of the setup and its optimized performance would provide a stronger foundation for future discussions on its practical applications.
Thanking for this useful suggestion, we believe the new version has made clear that main objective of the work is indeed related to the Raman setup optimization and testing.
Reviewer 2 Report
Comments and Suggestions for Authors
This manuscript developed a low-cost Raman spectrometer for medical use. This work is interesting,.
1. The performances of the model can be added in the abstract model.
2. The logic and the flowchart of introduction should be re-arranged.
3. Please summary the stability and robustness of the assembled system.
4. It might be better to compare the performances of the developed system and the well-known commercial instrument
5. In the introduction section, the authors indicated the potential use of the developed system for medical analysis. However, it seemed that only section 3 is for urine analysis, and no more information is explored.
6. The authors used the methanol-ethanol solutions to develop models. Why? It is quite confusing.
7. In the main text, which supervised learning model is used? Confusing.
8. How the supervised learning model is used for urine analysis?
Although this manuscript follows the general writing pattern, and the part for the system development is clear. It is unclear what are the objectives? Why such short information for section 3 for urine analysis? Why not use urine samples to develop the models? Why use the methanol-ethanol solutions to develop models? This manuscript is in chaos, and quite hard to follow.
Author Response
We thank the reviewer for the detailed review out for the manuscript. In this document, we report our answers to each comment, comments indicating the changes in the manuscript, also highlighted in yellow in the nmansucript. We hope to have improved the paper and made it more robust, comprehensive and easier to understand by addressing these comments.
- The performances of the model can be added in the abstract model.
As suggested, we added the performance metrics of the model in Abstract, lines 29-31.
- The logic and the flowchart of introduction should be re-arranged.
The introduction logic has been re-arranged.
- Please summary the stability and robustness of the assembled system.
The stability and robustness of the system was summarized in Section 5, lines 495 – 499.
- It might be better to compare the performances of the developed system and the well-known commercial instrument
In this study, we didn’t perform a comparison between our system and commercial Raman instruments, however we have added a paragraph explaining the most significant difference, its affordability, in Section 5, lines 511-514.
- In the introduction section, the authors indicated the potential use of the developed system for medical analysis. However, it seemed that only section 3 is for urine analysis, and no more information is explored.
The system indeed holds significant potential for diagnostics applications; however, the primary focus of this study was to develop and optimize a low-cost Raman spectroscopy system. The urine analysis presented is superficial, as it serves as a base to illustrate the system’s ability to acquire and detect Raman signals in this complex biofluid. A more comprehensive exploration of the clinical applications, including detailed urine analysis, is planned for future studies. This information was included in Section 5, lines 481 – 483 and lines 501 - 503.
- The authors used the methanol-ethanol solutions to develop models. Why? It is quite confusing.
We used methanol-ethanol solutions during the development of the supervised learning model because these solutions produce simpler Raman spectra, which facilitated the initial development and optimization of the algorithm. These simpler and well-known spectra, acquired by our Raman system, allowed us to evaluate the model’s ability to classify Raman spectra effectively before progressing to more complex samples, like urine. Furthermore, alcohol solutions are easier to handle and to obtain. We added this justification in Section 4.1.1, lines 403 – 407, for a better understanding.
- In the main text, which supervised learning model is used? Confusing.
The supervised learning model used in this study is a Convolutional Neural Network (CNN). We added this information in Section 4.1.1, lines 429 – 430.
- How the supervised learning model is used for urine analysis?
At this stage, the supervised learning model has not yet been applied to the urine analysis. However, the results obtained demonstrate that the model developed is able to classify Raman spectra, establishing a foundation for future work where the model can be trained with urine samples for diagnostic purposes.
Although this manuscript follows the general writing pattern, and the part for the system development is clear. It is unclear what are the objectives? Why such short information for section 3 for urine analysis? Why not use urine samples to develop the models? Why use the methanol-ethanol solutions to develop models? This manuscript is in chaos, and quite hard to follow.
The main objective of this study was the development of a portable and low-cost Raman spectroscopy-based system, capable of acquiring high-quality Raman spectra, for future diagnosis applications. Also, we wanted to evaluate the possibility of using a supervised learning model to classify Raman spectra. The objectives were added in Section 1, lines 93 – 102, for better understanding.
Round 2
Reviewer 1 Report
Comments and Suggestions for Authors
1. Please ensure the Authors wrote in the abstract ” The proposed system achieved 100% accuracy and precision, with a training time of approximately 5 minutes, demonstrating the model’s 30 potential for classifying simple Raman spectra under noisy conditions and paving the way for to 31 complex analysis.’’100 % accuracy is not possible (I am not satisfied with that, although the sample size is not enough). Maybe an overfitting issue occurs. If you use 100% data for training (for getting this result) then it does not mean you show those results.
2. Please remove the Matlab code, there is no need to provide the code.
3. The abstract and conclusion should have been linked.
Comments on the Quality of English LanguageThe quality of the English Language can be improved.
Author Response
We thank the reviewer for this additional useful advice. We fully followed her/his suggestion, as summarized below.
- Please ensure the Authors wrote in the abstract ” The proposed system achieved 100% accuracy and precision, with a training time of approximately 5 minutes, demonstrating the model’s 30 potential for classifying simple Raman spectra under noisy conditions and paving the way for to 31 complex analysis.’’100 % accuracy is not possible (I am not satisfied with that, although the sample size is not enough). Maybe an overfitting issue occurs. If you use 100% data for training (for getting this result) then it does not mean you show those results.
These results were obtained using 80% of the dataset for training, but they reflect the limitations of the small dataset size, which likely contributed to overfitting. We added this disclaimer in Section 5.1.3 lines 549-552 and altered the hyperparameters combination, Section 5.1.3 line 562 and changed the model performance in the Abstract.
- Please remove the Matlab code, there is no need to provide the code.
As suggested, the Matlab code was removed from Appendix A.
- The abstract and conclusion should have been linked.
We have revised the abstract for better correlation with the conclusions.
Reviewer 2 Report
Comments and Suggestions for Authors
The author failed to respond the comments except comment 1. The authors should carefully think what they want to present, and arrange the related content.
Author Response
We thank the reviewer for his deep analysis of the manuscript. We agree with his general vision about how to improve the workflow of the presentation. By rewriting, or reorganizing, many parts of the text, we have tried to make more evident the motivation of the work and how we plan to use our setup to support following work in biomedical sensing applications. The novel parts are highlighted in green and blue, while we kept the some of the yellow parts from the 1st round of revision. All the descriptions of the proposed Raman setup is organized in a manner to make clear to the reader which characteristics can be used in our applications, and which characteristics can be improved to better support its serviceability.
- The performances of the model can be added in the abstract model.
As suggested, we added the performance metrics of the model in Abstract
- The logic and the flowchart of introduction should be re-arranged.
The introduction logic has been re-arranged. The final goal of this setup is for a point-of-care device to be used for screening the risk of contracting AKI. So, we have introduced a more detailed description of the motivation for the work, including 4 new references. The present flow of the introduction is based on the following structure:
- the introduction to a problem (AKI risk screening)
- the proposed solution (portable Raman spectroscopy)
- the technological challenges to be addressed (laser quality and power, fluorescence noise and wavelength)
- The deep learning approach for overcoming the challenges and automatization of the classification
3. Please summary the stability and robustness of the assembled system.
Please see answer to point 4
4. It might be better to compare the performances of the developed system and the well-known commercial instrument
A discussion about resolution, stability and robustness, has been added in section 2. Lines 395-417, together with a comparison with other commercial portable raman systems.
5. In the introduction section, the authors indicated the potential use of the developed system for medical analysis. However, it seemed that only section 3 is for urine analysis, and no more information is explored.
The system indeed holds significant potential for diagnostics applications; however, the primary focus of this study was to develop and optimize a low-cost Raman spectroscopy system. The urine analysis presented is superficial, as it serves as a base to illustrate the system’s ability to acquire and detect Raman signals in this complex biofluid. A more comprehensive exploration of the clinical applications, including detailed urine analysis, is planned for future studies. This information was presented in a new section about data treatment and validation. Lines 449-512.
Also, to avoid confusion on the main subject that is the target of the manuscript, the title has been change, removing the reference to POC applications, keeping th focus of the Ramas prototype.
6. The authors used the methanol-ethanol solutions to develop models. Why? It is quite confusing.
We used methanol-ethanol solutions during the development of the supervised learning model because these solutions produce simpler Raman spectra, which facilitated the initial development and optimization of the algorithm. These simpler and well-known spectra, acquired by our Raman system, allowed us to evaluate the model’s ability to classify Raman spectra effectively before progressing to more complex samples, like urine. Furthermore, alcohol solutions are easier to handle and to obtain. We added this justification in Section 5.1.1, , for a better understanding. Some detail has been also added in the new section 4, including new references to support our approach. A table for the LOD we have obtained in our system, compared with data from other authors is also included.
- In the main text, which supervised learning model is used? Confusing.
The supervised learning model used in this study is a Convolutional Neural Network (CNN). The architecture includes a convolutional layer, batch normalization layer a ReLu activation function and a pooling layer. The output layers consist of a fully connected layer, followed by a SoftMax activation function and a classification layer. This information was added in Section 4.1.1, lines 428 – 438. A new Figure 16 was also added to better explain the model.
- How the supervised learning model is used for urine analysis?
At this stage, the supervised learning model has not yet been applied to the urine analysis. However, the results obtained demonstrate that the model developed is able to classify Raman spectra, establishing a foundation for future work where the model can be trained with urine samples for diagnostic purposes. A mention to a future clinical study has been included in the end of the new section 4.
Although this manuscript follows the general writing pattern, and the part for the system development is clear. It is unclear what are the objectives? Why such short information for section 3 for urine analysis? Why not use urine samples to develop the models? Why use the methanol-ethanol solutions to develop models? This manuscript is in chaos, and quite hard to follow.
We thank the reviewer for his deep evaluation of our manuscript and the useful advice for improving the quality of the text. We hope that with the present corrections and changes the goal of the work has been more clearly expressed. This is an important milestone in a larger long-term project. Having the Raman prototype operative and configures we may go further and request to the ethical committee the kickoff of a clinical study (that will last 2 years). Along this path we will be able to tune the machine learning algorithm over a clinical data set, and to optimize some detail of the Raman setup. This would not be possible to be done with a research level system, as the algorithm should be tuned for the specific machine we will use on the clinical data, so we decided to build the spectrometer as a first step. As a last future task, after the clinical validation, we plan to include an edge computing FPGA system to operate the spectra classification on a POC environment.